# LinkThief: Combining Generalized Structure Knowledge with Node Similarity for Link Stealing Attack against GNN

## ABSTRACT

Graph neural networks (GNNs) have a wide range of applications in multimedia. Recent studies have shown that GNNs are vulnerable to link stealing attacks, which infers the existence of edges in the target GNN's training graph. Existing methods are usually based on the *assumption* that links exist between two nodes that share similar posteriors; however, they fail to focus on links that do not hold under this assumption. To this end, we propose **LinkThief**, an improved link stealing attack that combines generalized structure knowledge with node similarity, in a scenario where the attackers' background knowledge contains partially leaked target graph and shadow graph. Specifically, to equip the attack model with insights into the link structure spanning both the shadow graph and the target graph, we introduce the idea of creating a Shadow-Target Bridge Graph and extracting edge subgraph structure features from it. Through theoretical analysis from the perspective of privacy theft, we first explore how to implement the aforementioned ideas. Building upon the findings, we design the Bridge Graph Generator to construct the Shadow-Target Bridge Graph. Then, the subgraph around the link is sampled by the Edge Subgraph Preparation Module. Finally, the Edge Structure Feature Extractor is designed to obtain generalized structure knowledge, which is combined with node similarity to form the features provided to the attack model. Extensive experiments validate the correctness of theoretical analysis and demonstrate that **LinkThief** still effectively steals links without extra assumptions.

## CCS CONCEPTS

• **Do Not Use This Code** → **Generate the Correct Terms for Your Paper**; *Generate the Correct Terms for Your Paper*; Generate the Correct Terms for Your Paper; Generate the Correct Terms for Your Paper.

## KEYWORDS

Graph Neural Networks, Link Stealing Attacks, Privacy Attacks

## 1 INTRODUCTION

Over the past few years, Graph Neural Networks (GNNs) have experienced significant development. Due to their excellent performance in modeling graph-structured data, GNNs have enabled a variety of successful multimedia applications, such as social network analysis [10, 27, 42], multimedia search [9, 41], and recommendation [35, 39, 47]. Despite their excellent performance in various tasks, recent studies have shown that graph neural networks are vulnerable to privacy attacks such as model extraction attacks [26, 49], property inference attacks [34, 48], and membership inference attacks [6, 16, 24]. In this paper, we focus on link stealing attack, which is a link-level membership inference attack aimed at inferring whether a specific link (edge) exists in the training graph of the target GNN model. After the model owner has deployed and published the GNN online model, attackers can launch the attack by querying the target GNNs (i.e., a black-box setting) to obtain the nodes' posteriors, which poses a risk of link privacy leakage. For example, in a GNN-based physician recommendation system [3, 23], the patient and the heart specialist are represented as two nodes in the graph. The attackers hijack the representations of two nodes and input them into the attack model to infer the existence of a link between the two nodes and then infer whether the patient has a heart disease. This triggers a trust crisis in GNN systems.

GNNs obtain the context of nodes through a message passing mechanism [11, 19]. This process results in neighbors having similar posteriors, which, in turn, reveals private relationships between nodes. All link stealing attacks [15, 36, 43] utilize the similarity between the posteriors of two nodes as features to train the attack model. However, this may not be applicable to all links. For instance, in a task where one predicts users' genders in a social network, users with different genders have different posteriors. Yet, in real-world social networks, it is common for users of opposite genders to follow each other. Therefore, we need additional information to guide the attack model to steal this type of link. The edge subgraph around the link contains sufficient neighbor information [21, 44, 53]. When the attack background knowledge includes partially leaked target graph and shadow graph, we can extract the edge subgraph around the link to obtain the link structure features.

However, the shadow graph is inherently different from the target graph, which inevitably leads to the structure shift and the covariate shift between the two graphs [20, 38]. Furthermore, the edges within the shadow and target graphs exhibit different neighbor structures, implying that the edge subgraph structure features may be distinct [29, 52]. To this end, we introduce the idea of constructing a Shadow-Target Bridge Graph and extracting the edge subgraph structure features from it. However, how to implement this idea is a non-trivial challenge. Fortunately, through theoretical analysis from the perspective of privacy theft, we propose the following: (1) The quality of bridge construction can be measured by computing the distributional distance between node posteriors from the shadow graph and those from the partial target graph within edge subgraphs. (2) When the edge subgraphs are sampled on the Shadow-Target Bridge Graph, having more nodes from the partial target graph proves to be more beneficial. This process helps guide the subgraph structure features of the shadow link towards those of the target link, thereby enhancing the attack model with additional structural knowledge similar to the target graph.

Building upon these insights, we propose LinkThief, an improved link stealing attack that combines generalized structure knowledge with node similarity. LinkThief consists of three key modules. The first module is the Bridge Graph Generator, which concatenates the partial target graph, shadow graph, and bridge learned through policy gradient method REINFORCE, thereby generating a

Shadow-Target Bridge Graph. The second module is the Edge Sub-graph Preparation Module, which adopts different edge subgraph sampling methods for shadow links and target links, and assigns distinct structure labels to nodes within them. The third module is the Edge Structure Feature Extractor, which acquires the edge subgraph structure features fused with implicit similarity through cross-view contrastive learning between the raw edge subgraph and the similarity-preserving subgraph. In this way, attackers obtain generalized edge subgraph structure features that span the shadow graph and the target graph. Finally, we concatenate the edge subgraph structure features with explicit node similarity to form the input features for the attack model, thereby obtaining the link stealing results. To summarize, the contribution of this paper is as follows:

- **Problem**: This paper focuses on how to steal links that are invulnerable to similarity-based attacks. We first empirically analyze the bottleneck of using only node similarity as attack features, and then propose the idea of complementing attack features with edge subgraph structural features sampled from bridge graphs.
- **Methodology**: Through theoretical analysis, we explore how to implement the aforementioned idea. On this basis, we propose LinkThief, an improved link stealing attack that comprises three modules to extract generalized structure features of edge subgraphs around links as supplementary for the attack model.
- **Evaluation**: Comprehensive experiments on real-world datasets demonstrate the effectiveness of LinkThief in stealing links where similarity-based attacks are ineffective.

## 2 PRELIMINARIES

### 2.1 Victim GNNs Model

Due to their high capability in modeling non-Euclidean structured data, GNNs play a crucial role in multimedia applications. Since privacy relationships between nodes are included in these GNN-based multimedia applications, they are vulnerable to privacy attacks.

GNNs leverage graph structures and node features to learn low-dimensional representations for each node, and then map these representations to labels. Mainstream GNNs [13, 19, 31, 40] currently follow the message passing mechanism. For example, in node classification tasks, GNNs aggregate rich information from higher-order neighbors by stacking multiple graph convolutional layers, and finally output node classification results in the form of probability distributions over a set of labels, which are commonly referred to as posterior probabilities. Due to the message passing mechanism, neighbors have similar posteriors, which, in turn, reveal private relationships between nodes.

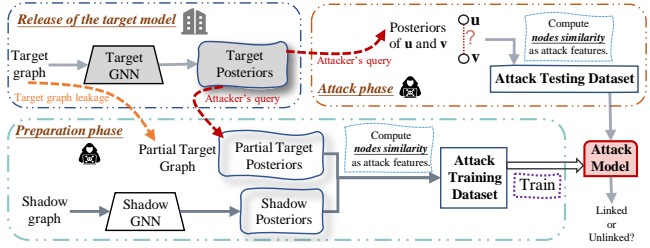

**Figure 1: The framework of Link Stealing Attacks.**

## 2.2 Threat Model

**Link Stealing Attack (LSA)** aims to infer the existence of attackers' targeted links in the training data of the target model. The vanilla LSA assumes three adversary's background knowledge: target dataset's nodes' features $\mathcal{X}^{tar}$, target dataset's partially leaked graph $\mathcal{A}^{tar\_leak}$, shadow dataset $\mathcal{G}^{sha}$. Whether the attackers possess each of these three items is a binary choice. Therefore, the attacker has eight different types of background knowledge, corresponding to eight different link stealing attacks. In this paper, we focus on knowledge $\mathcal{K} = (\mathcal{A}^{tar\_leak}, \mathcal{G}^{sha})$, which corresponds to LSA-4 in [15]. $\mathcal{K}$ is easily accessible in real-world settings. For instance, in a scenario where some user relationships of a multimedia social networking company have been disclosed, a rival company may use them as a partial target graph and leverage its own user social network as a shadow dataset to train a link stealing model aimed at trade secrets.

The attack pipeline is shown in Fig.1. Consider a GNN model designed for node classification as the target model that can be accessed by any user through black-box access. The output for a given node is a posterior vector, where the $i$-th probability represents the likelihood that the node belongs to the $i$-th class. During the preparation phase, attackers train a shadow GNN model to mimic the target GNN using the shadow dataset. After obtaining the partially leaked target graph, target model, and shadow model, attackers query them to obtain the posteriors of nodes in both the shadow graph and the partial target graph. Relying on the principle that similar nodes are more likely to be connected, attackers compute 12 posterior similarity metrics proposed in [15] for pairs of nodes. These 12 metrics constitute the attack features that are input into an attack model to predict the existence of links. During the attack phase, attackers query the posteriors of two nodes to be targeted using the target GNN model. The rest of the attack flow is the same as the preparation phase.

## 3 EXPLORATORY ANALYSIS

### 3.1 Notations

Let us denote a graph with $N$ nodes as $\mathcal{G} = (\mathcal{V}, \mathcal{E}, \mathcal{X}, \mathcal{Y})$, where $\mathcal{V} = \{v_1, v_2, \ldots, v_N\}$ is the set of nodes, $\mathcal{E} \in \mathcal{V} \times \mathcal{V}$ is the set of edges and $|\mathcal{E}|$ is the total number of edges, $\mathcal{X} = \{x_i\}_{N \times D}$ is a feature matrix with $D$-dimensional node feature vectors $x_i$, and $\mathcal{Y} = \{y_i\}_{N \times C}$ is a label matrix with $C$-dimensional node posterior probabilities $y_i$. $\mathcal{A} = \{a_{ij}\}_{N \times N}$ is an adjacency matrix($a_{ij}$=1 means the link between node $i$ and $j$ exists and 0 otherwise). We use $\mathcal{G}^{sha} = (\mathcal{V}^{sha}, \mathcal{E}^{sha}, \mathcal{X}^{sha}, \mathcal{Y}^{sha})$ to denote the shadow graph and $\mathcal{G}^{tar} = (\mathcal{V}^{tar}, \mathcal{E}^{tar}, \mathcal{X}^{tar}, \mathcal{Y}^{tar})$ to denote the target graph. Correspondingly, the adjacency matrix of $\mathcal{G}^{sha}$ and $\mathcal{G}^{tar}$ are denoted as $\mathcal{A}^{sha}$ and $\mathcal{A}^{tar}$, respectively. We refer to the partial target graph that the attacker is aware of as $\mathcal{G}^{tar\_leak} = (\mathcal{V}^{tar\_leak}, \mathcal{E}^{tar\_leak})$, while those unknown to the attacker are termed $\mathcal{G}^{tar\_safe} = (\mathcal{V}^{tar\_safe}, \mathcal{E}^{tar\_safe})$. Similarly, $\mathcal{A}^{tar} = \mathcal{A}^{tar\_leak} \cup \mathcal{A}^{tar\_safe}$.

### 3.2 Bottlenecks of Using Nodes Similarity as Attack Feature

Previous link stealing attacks [15, 36, 43] exploit similarities as attack features to train attack models under the assumption of

homogeneity, which has a high probability of success for most link stealing attacks. However, for some links, the posteriors of nodes may not be similar, which leads to the failure of similarity-based attack methods. For a pair of nodes $(v_i, v_j)$, the attacker aims to make a choice between the following two hypotheses:

- **Null hypothesis** $\mathbb{H}_0$: In the graph $\mathcal{G}$, there exists a link between nodes $v_i$ and $v_j$, that is, $e_{ij} = (v_i, v_j) \in \mathcal{E}$.
- **Alternative hypothesis** $\mathbb{H}_1$: In the graph $\mathcal{G}$, there exists no link between nodes $v_i$ and $v_j$, that is, $e_{ij} = (v_i, v_j) \notin \mathcal{E}$.

Given these two hypotheses $\mathbb{H}_0$ and $\mathbb{H}_1$, we use $\tilde{\mathbb{H}}_0$ and $\tilde{\mathbb{H}}_1$ to denote the attacker's predictions, where $\tilde{\mathbb{H}}_0$ represents the attacker accepts the null hypothesis, while $\tilde{\mathbb{H}}_1$ signifies the attacker accepts the alternative hypothesis. We categorize all links into four classes: True Positive(TP, truth is $\mathbb{H}_1$ and attacker accepts $\tilde{\mathbb{H}}_1$), False Negative(FN, truth is $\mathbb{H}_1$ yet attacker accepts $\tilde{\mathbb{H}}_0$), True Negative(TN, truth is $\mathbb{H}_0$ and attacker accepts $\tilde{\mathbb{H}}_0$), False Positive(FP, truth is $\mathbb{H}_0$ but attacker accepts $\tilde{\mathbb{H}}_1$). To further illustrate the bottleneck of relying solely on similarities as attack features, we visualize the attack features of these four types of links using a scatter plot generated by the t-SNE algorithm. Fig. 2 (a)(b) are from two real-world datasets, We observe that most links are successfully stolen, belonging to TP and TN, but there are still a small number of links that failed to be stolen, belonging to FN and FP. Edges classified as FN(FP) show considerable discrepancies in their attack features compared to those classified as TP(TN). This indicates that relying solely on similarity as the criterion for edge existence is insufficient for edges belonging to FN and FP.

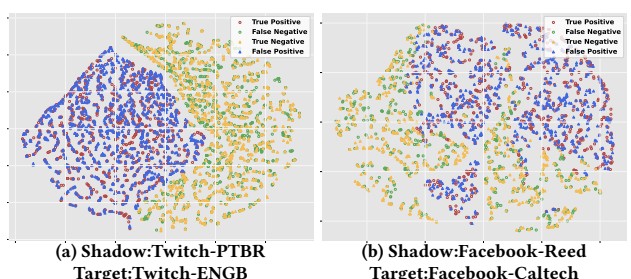

(a) Shadow:Twitch-PTBR
Target:Twitch-ENGB

(b) Shadow:Facebook-Reed
Target:Facebook-Caltech

**Figure 2: T-SNE visualization of attack features for links classified as TP, FP, TN, FN across two attack cases.**

### 3.3 Shadow-Target Bridge Graph

Given the bottleneck of attack models that depend solely on node similarities, we need additional link knowledge. The local enclosing subgraph around each link contains rich neighborhood structural information [44, 53]. Therefore, we can extract structural features from the edge subgraph as link knowledge, which serves as a supplement to attack features based on node similarity. We use $\mathcal{G}_{i,j}^r$ to denote the subgraph of edge $(v_i, v_j)$ within $r$ hops.

In practical scenarios, the number of leaked links is significantly fewer than those that remain safe in the target graph, i.e., $|\mathcal{E}^{tar\_leak}| \ll |\mathcal{E}^{tar\_safe}|$. This results in a small attack training dataset, making it challenging to capture comprehensive structure-aware edge subgraph representations in the target graph. To acquire universal and generalizable edge structural features, we introduce the shadow graph [24] that provides supplementary structural

knowledge to the attack model. However, the shadow graph is inherently different from the target graph, which inevitably leads to structure shift and covariate shift between the two graphs [20, 38]. Furthermore, the edges within the shadow and target graphs exhibit different neighbor structures, implying that the edge subgraph structure features may be distinct [29, 52]. Inspired by [4, 5], we try to build a bridge between the shadow graph and partial target graph to form the Shadow-Target Bridge Graph $\mathcal{G}^{st}$, the following is the definition:

*Definition 3.1 (Shadow-Target Bridge Graph).* The Shadow-Target Bridge Graph is represented as $\mathcal{G}^{st} = (\mathcal{V}^{st}, \mathcal{E}^{st}, \mathcal{X}^{st})$, where $\mathcal{V}^{st} = \mathcal{V}^{sha} \cup \mathcal{V}^{tar\_leak}$ is the nodes set, $\mathcal{X}^{st}$ is the feature matrix, and $\mathcal{E}^{st} = \mathcal{E}^{sha} \cup \mathcal{E}^{tar\_leak} \cup \mathcal{E}^{bridge}$ is the edges set where $\mathcal{E}^{bridge}$ serves as intermediaries connecting $\mathcal{G}^{sha}$ and $\mathcal{G}^{tar\_leak}$.

After querying the GNN model, the attacker obtains the nodes' posteriors. We use posteriors as node features $\mathcal{X}^{st}$. It is noteworthy that $\mathcal{X}^{st}$ does not equal the initial node features/attributes used in training for either the target or shadow GNN, because the attacker can only query the posteriors of nodes but cannot obtain their initial features. The Shadow-Target Bridge Graph defines the scope of knowledge transfer and distribution alignment under distribution shift between nodes, thereby forming a global perspective of the attack model. We can extract the edge subgraph $\mathcal{G}_{i,j}^r$ around the link $(v_i, v_j)$ on it, and input them into the edge structure feature extractor, which comprises GNN encoders, to derive link structure features. This approach treats the structure features as generalized knowledge across the shadow graph and target graph.

### 3.4 Theoretical Analysis of Privacy Theft

Although we propose addressing the acquisition of generalizable link structure knowledge by sampling edge subgraphs on the shadow-target bridge graph, we still confront two problems:

**RQ1:** How can the shadow graph be effectively connected to the partial target graph to construct the bridge graph?

**RQ2:** What strategy should be employed to sample neighbors around edges in order to construct edge subgraphs?

In the following discussion, we will analyze these two issues from the perspective of privacy theft.

*3.4.1 Problem Setup.* Since edge subgraph sampled from the bridge graph contains nodes from both partial target graph and shadow graph, we start with a formal definition of target nodes density:

*Definition 3.2.* [Density of target nodes in the edge subgraph]

$$D = \frac{1}{|\mathcal{V}_{i,j}^r|} \sum_{v \in \mathcal{V}_{i,j}^r} \frac{|\{u | u \in N_v^t\}|}{|\{u | u \in N_v^t\}| + |\{u | u \in N_v^s\}|}, \quad (1)$$

where $\mathcal{V}_{i,j}^r$ is the nodes set of the edge subgraph of $(v_i, v_j)$. $N_v^t$ is the neighbor set of node $v$ that belong to partial target graph, and $N_v^s$ is the neighbor set of node $v$ that belong to shadow graph. $D$ ranges from 0 to 1, with a larger $D$ indicating a higher proportion of nodes from partial target graph in the edge subgraph.

Different from [50], which uses the distribution distance of two different sensitive groups to measure the privacy leakage of message passing, we define the privacy theft of edge subgraph structure feature extraction process as follows:

*Definition 3.3 (Measurement of privacy theft for edge subgraph structure feature extraction).* Given the link and its corresponding edge subgraph, the nodes' features and representations after edge subgraph structure feature extraction follow the distributions $\mathbf{P}$ and $\widetilde{\mathbf{P}}$, respectively. Privacy theft for edge subgraph structure feature extraction is measured by the distance between $\mathbf{P}$ and $\widetilde{\mathbf{P}}$.

For the convenience of subsequent analysis, we employ the Wasserstein distance to measure the distance between two multivariate normal distributions $p \sim \mathcal{N}(\mu_p, \Sigma_p)$ and $q \sim \mathcal{N}(\mu_q, \Sigma_q)$:

$$\mathcal{W}[p,q] = (\mu_p - \mu_q)^T(\mu_p - \mu_q) + \mathrm{Tr}(\Sigma_q) - 2\mathrm{Tr}((\Sigma_p^{\frac{1}{2}}\Sigma_q\Sigma_p^{\frac{1}{2}})^{\frac{1}{2}}). \quad (2)$$

Intuitively, privacy theft in structure feature extraction can be understood as the extent to which nodes extract features from their neighbors. When the distributions between features and learned representations are farther apart, it means that each node in the edge subgraph extracts more context from its neighbors. The sum of these extractions represents the privacy theft of the edge subgraph structure feature extraction. Larger privacy theft means that the attacker obtains more structural information about the link. Therefore, it is easier to infer the existence of a connection.

*3.4.2 Theoretical Analysis.* Contextual Stochastic Block Model (CSBM) [8, 22] is a random graph model that adeptly combines graph structure with node features, enabling the effective simulation of graphs with community structures. Since the edge subgraph includes nodes from both the partial target graph and shadow graph, we employ CSBM for our analysis, treating the edge subgraph with $n$ nodes as a random graph $\mathcal{G}_{i,j}^r \sim (n, p, q, \mu, k\mu, d)$. Each node is associated with a community label: $v^t$ represent the target nodes and $v^s$ represent the shadow nodes. We construct edges based on two types of probabilities: if a node is linked to $v^t$, an edge between them is generated with a probability of $p$; otherwise, if it is linked to $v^s$, the probability of $q$. Based on community labels, $d$-dimensional feature vectors $x$ are sampled differently: we denote feature vectors of $v^t$ as $x^t$, each dimension of $x^t$ follows $\mathcal{N}(\mu, 1)$, whereas those of $v^s$ as $x^s$, each dimension of $x^s$ follows $\mathcal{N}(k\mu, 1)$. Thus, $x^t$ follow the normal distribution $\mathcal{N}(\mu_{x^t}, \Sigma_{x^t})$ and $x^s$ follow the normal distribution $\mathcal{N}(\mu_{x^s}, \Sigma_{x^s})$, where

$$\mu_{x^t}[i] = \mu, \; \mu_{x^s}[i] = k\mu, \; \Sigma_{x^t}[i,i] = \Sigma_{x^s}[i,i] = 1 \; (0 \le i < d). \quad (3)$$

We will analyze privacy theft in the process of extracting edge subgraph structure features from two different scenarios.

**Best-case scenario.** The ideal edge subgraph of an edge consists exclusively of nodes $v^t$. Considering a 1-layer GCN model without nonlinearity, a standard message passing $\mathbf{Z} = \tilde{\mathbf{A}}X$ where $\tilde{\mathbf{A}}$ is the normalized adjacency matrix with self-loop, the representation of node $a$ after propagation can be written as:

$$z_a = \frac{1}{|N_a|}x_a^t + \sum_{b \in N_a^t} \frac{1}{\sqrt{|N_a||N_b|}}x_b^t, \quad (4)$$

where $N_a$ denotes the neighbor set of node $v_a$. Consider the generation process of the synthetic edge subgraph that only includes nodes $v^t$, which means $N_a = N_a^t$. For each node, the approximate size of its neighbor set can be expressed as $np$. Thus, representations of nodes follow distributions $\mathbf{z} \sim \mathcal{N}(\mu_z, \Sigma_z)$, where

$$\mu_z[i] = \frac{1+np}{np}\mu, \quad \Sigma_z[i,i] = \frac{1+np}{n^2p^2} \quad (0 \le i < d). \quad (5)$$

Using Eq.2 and Def.3.3, in the optimal case where the edge graph only includes $v^t$, privacy theft ($PT$) of edge subgraph structure feature extraction can be quantified as:

$$PT^{opt} = d\mu^2(\frac{1}{np})^2 + d\left(\sqrt{\frac{np+1}{n^2p^2}} - 1\right)^2. \quad (6)$$

**General-case scenario.** Under general circumstances, the edge subgraph of edge is composed of nodes $v^t$ and $v^s$. This inevitably introduces noise into $v^t$ in the form of node features and structures originating from $v^s$, thereby leading to covariate shift and structure shift within the edge graph. After one layer of graph convolution, the representation of node $v_a$ can be written as:

$$z_a = \frac{1}{|N_a|}x_a + \sum_{b \in N_a^t} \frac{1}{\sqrt{|N_a||N_b|}}x_b^t + \sum_{b \in N_a^s} \frac{1}{\sqrt{|N_a||N_b|}}x_b^s, \quad (7)$$

where $N_a$ denotes the neighbor set of node $v_a$. Each node's neighbors consist of both nodes $v^t$ and $v^s$, hence $N_a = N_a^t \cup N_a^s$. The size of the neighbor set can be approximately represented as $n(p+q)$. A percentage of $p/(p+q)$ of its neighbors are $v^t$, whereas a percentage of $q/(p+q)$ of its neighbors are $v^s$. The representations $z^t$ of nodes $v^t$ follow $\mathcal{N}(\mu_{z^t}, \Sigma_{z^t})$, while $z^s$ of nodes $v^s$ follow $\mathcal{N}(\mu_{z^s}, \Sigma_{z^s})$. To facilitate analysis, we combine $z^t$ and $z^s$, such that the representations $z$ of all nodes in the edge subgraph approximately follow $\mathcal{N}(\mu_z, \Sigma_z) = \mathcal{N}((\mu_{z^t} + \mu_{z^s})/2, (\Sigma_{z^t} + \Sigma_{z^s})/2)$. Correspondingly, initial feature vectors $x$ approximately follow $\mathcal{N}(\mu_x, \Sigma_x) = \mathcal{N}((\mu_{x^t} + \mu_{x^s})/2, (\Sigma_{x^t} + \Sigma_{x^s})/2)$, where

$$\mu_z[i] = \frac{(k+1) + 2n(p+kq)}{2n(p+q)}\mu, \quad \sigma_z[i,i] = \frac{n(p+q)+1}{n^2(p+q)^2},$$
$$\mu_x[i] = \frac{(k+1)}{2}\mu, \quad \sigma_z[i,i] = 1. \quad (8)$$

Using Eq.2 and Def.3.3, in the general case where the edge subgraph contains both $v^t$ and $v^s$, privacy theft of edge subgraph structure feature extraction can be quantified as:

$$PT = d\mu^2[\frac{1}{n(p+q)}\frac{1+k}{2} + \frac{p-q}{p+q}\frac{1-k}{2}]^2 + d\left(\sqrt{\frac{n(p+q)+1}{n^2(p+q)^2}} - 1\right)^2. \quad (9)$$

**Further analysis.** In the above discussion, two distinct scenarios of privacy theft were derived based on both the optimal scenario and the general scenario. To approximate the level of privacy theft in Eq.9 to that of Eq.6, i.e., to make $\Delta PT = PT^{opt} - PT$ approach 0 as much as possible, it can be readily observed that $k$ should equal 1. Then, we substitute $D = p/(p+q)$ defined in definition(3.2) into $\Delta PT$, we can transform it into the following form:

$$\Delta PT = d[\frac{1+D}{n^2p^2}\mu^2 + \frac{np+1+D}{np(\sqrt{np+1} + \sqrt{npD+D^2})}$$
$$\cdot \frac{\sqrt{np+1} + \sqrt{npD+D^2} - 2np}{np}](1-D). \quad (10)$$

See the appendix for the detailed derivation. We can then have the following propositions:

**PROPOSITION 3.4.** *Given $\mathcal{G}_{i,j}^r \sim (n, p, q, \mu, k\mu, d)$ and $D = \frac{p}{p+q}$,*

(1) *As $k$ approaches 1, $\Delta PT$ tends towards 0. This implies that the closer the features of $v^s$ are to those of $v^t$, the closer $PT$ approaches its optimal level.*

(2) *As D approaches 1, $\Delta PT$ tends towards 0. This implies that in the edge subgraph, the larger the proportion of nodes $v^t$ among all nodes , the closer PT approaches its optimal level.*

These propositions answer **RQ1** and **RQ2**, suggesting that the more similar the features between $v^t$ and $v^s$ and the higher the proportion of $v^t$, the more the subgraph structure around links can be stolen during the extraction of edge subgraph structure feature. In other words, this provides the attacker with more structure-aware knowledge about links that are targets for attack.

## 4 THE PROPOSED METHOD

In this section, we detail the design of the proposed link stealing attack framework — LinkThief, which combines generalized link structure knowledge with node similarity.

### 4.1 Overview

We have partially leaked target graph (containing only the leaked links) and complete shadow graph. The target model is a black box model (i.e., the adversary can only access the node posterior/embedding without knowing the model's parameters), while the shadow model is a white box model that we have trained using the shadow graph. We query the posteriors of target nodes and shadow nodes from the target model and shadow model, and use them as new features $\mathcal{X}^{tar\_leak}$ and $\mathcal{X}^{sha}$, respectively. According to Prop.3.4, we design **B**ridge **G**raph **G**enerator (BGG) for **RQ1** and **E**dge **S**ubgraph **P**reparation **M**odule (ESPM) for **RQ2**, to construct the Shadow-Target Bridge Graph and sample edge graphs from it, respectively. On this basis, **E**dge **S**tructure **F**eature **E**xtractor (ESFE) is proposed to learn generalized subgraph structure feature across these two graphs. As defined in [15], a set of 12 distance metrics quantifying the similarity between two vectors constitutes the node similarity features. Finally, we concatenate the edge subgraph structure features with node similarity features to form the attack features. These features are then input into an attack model consisting of Multi-Layer Perceptron (MLP) to derive inference results. In the attack testing phase, the attack flow is the same as above. Although our framework is modularized in LinkThief, each component is intertwined to learn the link features. The overall framework is illustrated in the top left corner of Fig. 3. The subsequent chapters will individually introduce the aforementioned three proposed modules.

### 4.2 Bridge Graph Generator (BGG)

Considering the shadow graph with $M$ nodes and the partial target graph with $N$ nodes, bridges refer to a collection of inter-graph links connecting nodes $v^s$ to nodes $v^t$. The bridge learner consists of a parametric matrix $\omega = \{w_{mn}\}_{M \times N}$. The probability of adding an edge between the node $v_m$ and $v_n$ is $p(a_{mn}) = \frac{\exp(w_{mn})}{\sum_{n'} \exp(w_{mn'})}$. The set $\{b_{mn_t}\}_{t=1}^{S}$ with $S$ edges is sampled from a multinomial distribution $\mathcal{M}(p(a_{m1}), \cdots, p(a_{mN}))$, which give the nonzero entries in the $m$-th row of $\mathcal{A}^{bridge}$. We concatenate $\mathcal{E}^{bridge}$ corresponding to $\mathcal{A}^{bridge}$ with $\mathcal{E}^{sha}$ and $\mathcal{E}^{tar\_leak}$ to get $\mathcal{E}^{st}$, and concatenate the posteriors queried from the target and shadow model to obtain nodes features $\mathcal{X}^{st}$. Finally, They are fed into the GNN encoder with

parameters $\phi$, which yields the representations $Z^{st}$. $Z^{st}$ consists of $Z^s$, representations of $v^s$, and $Z^t$, representations of $v^t$.

Regarding **RQ1**, the Prop.3.4(1) presents a criterion to evaluate the effectiveness of bridges. Namely, the closer between $Z^t$ and $Z^s$, the more bridges facilitate subsequent privacy theft. Moreover, since our original intention is to serve the shadow graph as a supplement to target graph, we aim to ensure that $Z^{st}$ cannot deviate too far from the feature $\mathcal{X}^{tar\_leak}$. We define the above two distances as $\mathcal{L}_{inner}$ and $\mathcal{L}_{outer}$ respectively, which can be expressed as:

$$\mathcal{L}_{inner} = \mathcal{W}(Z^s, Z^t), \ \mathcal{L}_{outer} = \mathcal{W}(\mathcal{X}^{tar\_leak}, Z^{st}). \quad (11)$$

$\mathcal{W}$ is the Wasserstein-1 distance [2, 12], which we use to measure the distribution distance similar to the theoretical analysis:

$$\mathcal{W}(\mathbb{P}, \mathbb{Q}) = \inf_{\gamma \in \Pi(\mathbb{P}, \mathbb{Q})} \mathbb{E}_{(z,z') \sim \gamma} \left[ \| z - z' \| \right], \quad (12)$$

where $z$ and $z'$ are two random variables sampled from two different distributions $\mathbb{P}$ and $\mathbb{Q}$ separately. Due to the high computational complexity of the original Wasserstein distance, we use the Sinkhorn algorithm [7] to efficiently approximate it through an iterative normalization procedure.

The optimization for parametric $\omega$ is difficult because the edge sampling process is non-differentiable and hinders back-propagation. To handle it, we use policy gradient method REINFORCE [1, 38, 51], treating edge generation as a decision process and edge adding as actions. Specifically, We use $-\mathcal{L}_{inner}$ as the reward function $R(\mathcal{A}^{bridge})$, i.e. the smaller the distance between $Z^s$ and $Z^t$, the greater the reward. The bridge learner's $\omega$ and GNN's $\phi$ are updated as follows:

$$\omega \leftarrow \omega + \eta \nabla_{\omega} \log p_{\omega}(\mathcal{A}^{bridge}) R(\mathcal{A}^{bridge}), \quad (13)$$

$$\phi \leftarrow \phi - \eta \nabla_{\phi}(\mathcal{L}_{inner} + \mathcal{L}_{outer}), \quad (14)$$

where $p_{\omega}(\mathcal{A}^{bridge}) = \Pi_{m=1}^{M} \Pi_{t=1}^{S} p(b_{mn_t})$, $\eta$ are the learning rates.

### 4.3 Edge Subgraph Preparation Module (ESPM)

Regarding **RQ2**, the Prop.3.4(2) suggests that the higher the density of $v^t$ in edge subgraph, the more conducive it is for privacy theft. Therefore, in the edge subgraph sampler, the sampling methods for target links and shadow links are different. Specifically, When sampling the edge subgraph in the Shadow-Target Bridge Graph, for target links, we select neighbors only from $v^t$, whereas for shadow links, we choose neighbors not only from $v^s$ but also from $v^t$. This method guides the structure distribution of the shadow edge subgraph to approximate that of the target edge subgraph. Inspired by [44, 46], in order to mark nodes with different roles, we use Double-Radius Node Labeling(DRNL) method to assign structure labels to each node in the edge subgraph. For a node $v_x$ in the edge subgraph:

$$NL(x) = 1 + \min(d_i, d_j) + (d_{ij}/2)[(d_{ij}/2) + (d_{ij}\%2) - 1], \quad (15)$$

where $d$ denotes the shortest path distance between two nodes, $d_i = d(v_i, v_x), d_j = d(v_j, v_x), d_{ij} = d_i + d_j$. After getting structure labels, we concatenate their one-hot vectors with $\mathcal{X}^{st}$ to construct new node features of the edge subgraph.

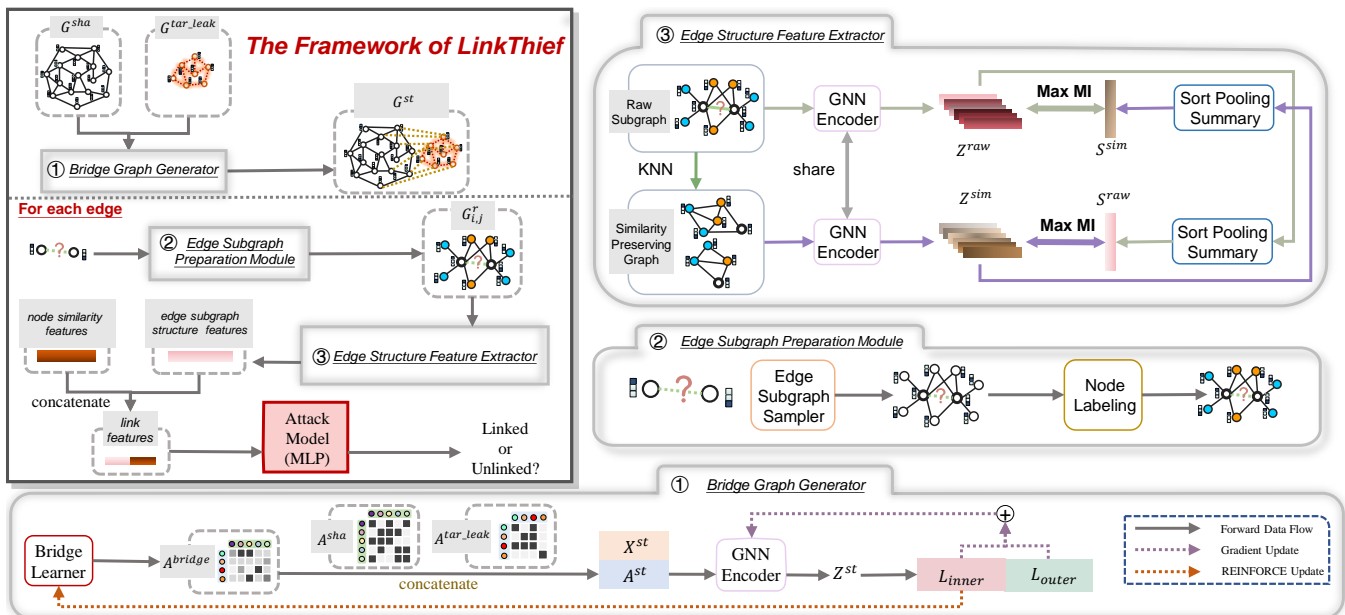

**Figure 3: The top left corner is the framework of LinkThief, surrounded by the three modules used in LinkThief.**

## 4.4 Edge Structure Feature Extractor (ESFE)

Further, we construct the $k$-NN graph [18] to capture latent relationships in the feature space, named the similarity-preserving graph. To ensure that each node in the edge subgraph contains implicit node similarity knowledge, we conduct cross-view contrastive learning between the raw and similarity-preserving graphs [14, 28]. In practice, we leverage a GNN encoder with parameters $\theta$ to obtain node representations of two views, denoted as $Z^{raw}$ and $Z^{sim}$. To extract the subgraph features, we use sort pooling [45] as the readout to obtain subgraph-level representations $S^{raw}$ and $S^{sim}$ of $Z^{raw}$ and $Z^{sim}$, respectively. To ensure that $Z^{raw}$ effectively captures the implicit node similarities, while $Z^{sim}$ retains the raw structure information, we maximize the mutual information (MI) between them. MI [17, 32] is widely used to measure the dependence between two distributions, which can be defined as:

$$\mathcal{I}(Z,S) = \frac{1}{2N}\left(\sum_{i=1}^{N}\log \mathcal{T}_{\psi}(Z_i,S) + \sum_{i=1}^{N}\log[1-\mathcal{T}_{\psi}(\tilde{Z}_i,S)]\right), \quad (16)$$

where $N$ is the number of nodes in the subgraph, $\mathcal{T}_{\psi}$ denotes an MI estimator composed of the Bilinear layer that provides probability scores for sampled pairs, $\tilde{Z}$ represents perturbed node embeddings as negative samples. Thus, our contrastive loss is defined as:

$$\mathcal{L}_{MI} = \mathcal{I}(Z^{raw}, S^{sim}) + \mathcal{I}(Z^{sim}, S^{raw}), \quad (17)$$

GNN's $\theta$ and MI estimator's $\psi$ are updated as follows:

$$\psi \leftarrow \psi + \eta\nabla_{\psi}\mathcal{L}_{MI}, \quad \theta \leftarrow \theta + \eta\nabla_{\theta}\mathcal{L}_{MI}. \quad (18)$$

## 5 EXPERIMENT

In this section, we first evaluate the effectiveness of LinkThief. Then, we explore the role of the three modules proposed by LinkThif. Finally, we empirically verify how Prop.3.4 affects the privacy theft of edge subgraph structure feature extraction.

## 5.1 Experimental Setup

**Datasets.** We use four real datasets from different domains including sixteen graphs for evaluation. The **Twitch dataset** [25] contains social networks from five regions (**ENGB, ES, TW, RU, PTBR**). The **Facebook dataset** [30] contains social networks from five US universities (**Caltech, Haverford, Reed, Simmons, Swarthmore**). The **ArnetMiner dataset** [33] contains citation networks from three academic databases (**DBLPv7, Citationv1, and ACMv9**). The **Airport dataset** [37] contains airport networks from three countries or regions (**Brazil, USA, and Europe**). The statistics of the datasets are given in the Appendix.

**Baselines.** Link stealing attacks are categorized into eight types based on the background knowledge of the attacker. Unlike other variants, vanilla LSA-4 in [15] is the closest to our setting when the attacker's knowledge includes partial target graphs and shadow datasets. In addition, we also investigate whether the shadow dataset provides additional link knowledge to the attack model, corresponding to LSA-3 in [15], where the attacker's background knowledge only includes the partial target graph. Therefore, we choose LSA-3 and LSA-4 as baselines.

**Models.** We choose GCN [19] as the target model and shadow model architecture. For a fair comparison, the hyperparameters are the same as in the previous work [15]. We carry out the experiments with the target graph leakage rate of 10%, 20%, and 30%, and list results with leakage rates of 10% and 20% in the Appendix. To simulate the attack, we generate a separate attack model for each pair of target and shadow datasets. So we construct an attack dataset (for training, validation, and testing) that comprises pairs of nodes and labels indicating whether they are linked. During the attack preparation phase, we consider all links from the partial target graph and shadow graph as positive samples, and select an equal number of unlinked node pairs as negative samples. We divide the above samples into the attack training/validation dataset at a 7:3 ratio.

**Table 1: Comparison of Ours and general link stealing attacks on Twitch dataset containing five social networks.**

| Target Dataset | Attack Method | Shadow Dataset | | | | | | | | | |
|---|---|---|---|---|---|---|---|---|---|---|---|
| | | ENGB | | ES | | PTBR | | RU | | TW | |
| | | ASR | AUC | ASR | AUC | ASR | AUC | ASR | AUC | ASR | AUC |
| ENGB | LSA-3 | - | - | 0.5415 | 0.5653 | 0.5320 | 0.5479 | 0.5405 | 0.5655 | 0.5327 | 0.5453 |
| | LSA-4 | - | - | 0.5002 | 0.5398 | 0.5238 | 0.5201 | 0.5204 | 0.5344 | 0.5214 | 0.5217 |
| | Ours | - | - | 0.7868 | 0.8609 | 0.7892 | 0.8639 | 0.7874 | 0.8615 | 0.7859 | 0.8599 |
| ES | LSA-3 | 0.5393 | 0.5793 | - | - | 0.5504 | 0.5811 | 0.5526 | 0.5819 | 0.5501 | 0.5786 |
| | LSA-4 | 0.5024 | 0.5078 | - | - | 0.5022 | 0.5043 | 0.4987 | 0.4993 | 0.5024 | 0.5018 |
| | Ours | 0.8347 | 0.9063 | - | - | 0.8327 | 0.9081 | 0.8294 | 0.9024 | 0.8303 | 0.9018 |
| PTBR | LSA-3 | 0.5509 | 0.5816 | 0.5594 | 0.5856 | - | - | 0.5571 | 0.5863 | 0.5457 | 0.5849 |
| | LSA-4 | 0.5134 | 0.5218 | 0.5000 | 0.5282 | - | - | 0.5130 | 0.5055 | 0.5218 | 0.5187 |
| | Ours | 0.8276 | 0.9048 | 0.8275 | 0.9034 | - | - | 0.8282 | 0.9049 | 0.8239 | 0.8994 |
| RU | LSA-3 | 0.5425 | 0.5622 | 0.5330 | 0.5577 | 0.5369 | 0.5553 | - | - | 0.5282 | 0.5412 |
| | LSA-4 | 0.5112 | 0.5055 | 0.5053 | 0.5260 | 0.4934 | 0.5023 | - | - | 0.4982 | 0.5009 |
| | Ours | 0.8217 | 0.8935 | 0.8139 | 0.8860 | 0.8182 | 0.8910 | - | - | 0.8167 | 0.8901 |
| TW | LSA-3 | 0.5287 | 0.5428 | 0.5316 | 0.5411 | 0.5234 | 0.5332 | 0.5278 | 0.5515 | - | - |
| | LSA-4 | 0.5136 | 0.5140 | 0.5003 | 0.5063 | 0.5173 | 0.5254 | 0.5128 | 0.5101 | - | - |
| | Ours | 0.8402 | 0.9146 | 0.8369 | 0.9106 | 0.8394 | 0.9112 | 0.8393 | 0.9127 | - | - |

**Table 2: Comparison of Ours and general link stealing attacks on Facebook dataset containing five social networks.**

| Target Dataset | Attack Method | Shadow Dataset | | | | | | | | | |
|---|---|---|---|---|---|---|---|---|---|---|---|
| | | Caltech | | Haverford | | Reed | | Simmons | | Swarthmore | |
| | | ASR | AUC | ASR | AUC | ASR | AUC | ASR | AUC | ASR | AUC |
| Caltech | LSA-3 | - | - | 0.5741 | 0.6193 | 0.5734 | 0.6190 | 0.5750 | 0.6160 | 0.5747 | 0.6173 |
| | LSA-4 | - | - | 0.5002 | 0.5398 | 0.5238 | 0.5201 | 0.5204 | 0.5344 | 0.5214 | 0.5217 |
| | Ours | - | - | 0.8073 | 0.8815 | 0.8099 | 0.8834 | 0.8050 | 0.8803 | 0.8027 | 0.8766 |
| Haverford | LSA-3 | 0.5767 | 0.6031 | - | - | 0.5766 | 0.6059 | 0.5727 | 0.6009 | 0.5798 | 0.6060 |
| | LSA-4 | 0.5024 | 0.5078 | - | - | 0.5022 | 0.5043 | 0.4987 | 0.4993 | 0.5024 | 0.5018 |
| | Ours | 0.8004 | 0.8801 | - | - | 0.8839 | 0.8842 | 0.8053 | 0.8818 | 0.8043 | 0.8834 |
| Reed | LSA-3 | 0.5464 | 0.5745 | 0.5476 | 0.5746 | - | - | 0.5484 | 0.5752 | 0.5442 | 0.5738 |
| | LSA-4 | 0.5134 | 0.5218 | 0.5000 | 0.5282 | - | - | 0.5130 | 0.5055 | 0.5218 | 0.5187 |
| | Ours | 0.7773 | 0.8532 | 0.7744 | 0.8456 | - | - | 0.7748 | 0.8522 | 0.7759 | 0.8489 |
| Simmons | LSA-3 | 0.5733 | 0.6115 | 0.5710 | 0.6083 | 0.5702 | 0.6126 | - | - | 0.5747 | 0.6136 |
| | LSA-4 | 0.5112 | 0.5055 | 0.5053 | 0.5260 | 0.4934 | 0.5023 | - | - | 0.4982 | 0.5009 |
| | Ours | 0.8134 | 0.8897 | 0.8145 | 0.8899 | 0.8132 | 0.8893 | - | - | 0.8143 | 0.8895 |
| Swarthmore | LSA-3 | 0.5727 | 0.5990 | 0.5729 | 0.5983 | 0.5704 | 0.5987 | 0.5731 | 0.6009 | - | - |
| | LSA-4 | 0.5136 | 0.5140 | 0.5003 | 0.5063 | 0.5173 | 0.5254 | 0.5128 | 0.5101 | - | - |
| | Ours | 0.8065 | 0.8857 | 0.8039 | 0.8839 | 0.8040 | 0.8861 | 0.8065 | 0.8868 | - | - |

During the attack phase, to create the attack testing dataset, we consider the unleaked links in the target graph as positive samples and choose an equal number of unlinked node pairs as negative samples. In the Bridge Graph Generator, we adopt 2-layer GCN as the GNN encoder. In the Edge Structure Feature Extractor, we adopt DGCNN [44, 45] as the GNN encoder, which is often used to extract subgraph features. In line with previous works [15], we use 3-layer MLP for the attack model.

**Metrics of Attacking.** We use the AUC (Area Under the ROC Curve) and ASR (Attack Success Rate) metrics to evaluate attacking performance, which is consistent with the recent work [43]. We independently run 5 times and report the mean result.

## 5.2 Main Experiments

We conduct the main experiment in a setting with 10 bridges connecting a shadow node to a target node. As shown in Table 1 and Table 2, LSA-3 and LSA-4 exhibit poor performance on the Twitch dataset and Facebook dataset, with ASR and AUC scores ranging between 0.5 and 0.6. These scores are marginally better than random guessing. This shows that the two datasets are typically insensitive to similarity-based attacks. However, the performance of LinkThief far exceeds those of LSA-3 and LSA-4. This indicates that LinkThief effectively steals links compared to similarity-based attacks. For example, in the Twitch dataset, when the target dataset is TW and the shadow dataset is ENGB, LinkThief improves the AUC score by 40% compared to LSA-4. As shown in Table 3 and Table 4, in

**Table 3: Comparison of Ours and general link stealing attacks on ArnetMiner dataset containing three citation networks.**

| Target Dataset | Attack Method | Shadow Dataset | | | | | |
|---|---|---|---|---|---|---|---|
| | | Dblpv7 | | Acmv9 | | Citationv1 | |
| | | ASR | AUC | ASR | AUC | ASR | AUC |
| Dblpv7 | LSA-3 | - | - | 0.8283 | 0.8969 | 0.8346 | 0.9001 |
| | LSA-4 | - | - | 0.8129 | 0.8605 | 0.8156 | 0.8658 |
| | Ours | - | - | 0.8313 | 0.9067 | 0.8378 | 0.9077 |
| Acmv9 | LSA-3 | 0.8321 | 0.8947 | - | - | 0.8402 | 0.9114 |
| | LSA-4 | 0.8049 | 0.8698 | - | - | 0.8262 | 0.8930 |
| | Ours | 0.8417 | 0.9092 | - | - | 0.8465 | 0.9226 |
| Citationv1 | LSA-3 | 0.8386 | 0.9018 | 0.8403 | 0.9147 | - | - |
| | LSA-4 | 0.8269 | 0.8762 | 0.8379 | 0.8855 | - | - |
| | Ours | 0.8470 | 0.9159 | 0.8498 | 0.9214 | - | - |

**Table 4: Comparison of Ours and general link stealing attacks on Airport dataset containing three airport networks.**

| Target Dataset | Attack Method | Shadow Dataset | | | | | |
|---|---|---|---|---|---|---|---|
| | | Brazil | | Europe | | USA | |
| | | ASR | AUC | ASR | AUC | ASR | AUC |
| Brazil | LSA-3 | - | - | 0.8152 | 0.8902 | 0.7956 | 0.8890 |
| | LSA-4 | - | - | 0.7286 | 0.8060 | 0.7292 | 0.7815 |
| | Ours | - | - | 0.8130 | 0.8881 | 0.7935 | 0.8828 |
| Europe | LSA-3 | 0.8293 | 0.9001 | - | - | 0.8264 | 0.8967 |
| | LSA-4 | 0.8015 | 0.8744 | - | - | 0.7699 | 0.8486 |
| | Ours | 0.8381 | 0.9038 | - | - | 0.8307 | 0.9013 |
| USA | LSA-3 | 0.8738 | 0.9413 | 0.8788 | 0.9380 | - | - |
| | LSA-4 | 0.8615 | 0.9273 | 0.8414 | 0.9156 | - | - |
| | Ours | 0.8917 | 0.9522 | 0.8871 | 0.9473 | - | - |

the ArnetMiner dataset and the Airport dataset, although LSA-3 and LSA-4 show better attack performance compared to the first two datasets, LinkThief still outperforms them nonetheless. Similarly, when the target dataset is Acmv9 and the shadow dataset is Dblpv7, LinkThief outperforms LSA-4 by 4%. Compared with LSA-3 and LSA-4, which steal links vulnerable to similarity-based attacks, LinkThief also has a considerable improvement in attack effectiveness.

In addition, from the four tables, we observe that LSA-3 consistently outperforms LSA-4. This suggests that the shadow dataset added to the vanilla LSA actually reverses the attack model's performance, contradicting the initial purpose of enhancing the attack model with additional background knowledge through the shadow dataset. But LinkThief basically outperforms LSA-3 on four datasets, indicating that LinkThief effectively incorporates additional structural knowledge from shadow datasets into the attack model.

## 5.3 Ablation Study

We conduct the ablation study to show the effectiveness of each component in our LinkThief as shown in Table 5. We design three LinkThief variants for analysis: (1)**w/o BGG**: A variant without the Bridge Graph Generator. (2)**w/o ESPM**: A variant without the Edge Subgraph Preparation Module. (3)**w/o ESFE**: A variant without the Edge Structure Feature Extractor.

**Impact of Bridge Graph Generator:** We find that without BGG, LinkThief decreases AUC scores by 1% to 2% on Citation dataset, 0.2% to 1% on airport dataset, and even 3% in some cases. This suggests that constructing the bridge graph benefits the attack model by providing a perspective that spans the shadow and target graphs to the link.

**Table 5: AUC comparison of Ours and its variants on Citation dataset and Airport dataset.**

| Target Dataset | | Dblpv7 | Dblpv7 | Acmv9 | Acmv9 | Citationv1 | Citationv1 |
|---|---|---|---|---|---|---|---|
| **Shadow Dataset** | | Acmv9 | Citationv1 | Dblpv7 | Citationv1 | Dblpv7 | Acmv9 |
| **Method** | w/o BGG | 0.8853 | 0.8872 | 0.8977 | 0.9137 | 0.9042 | 0.9153 |
| | w/o ESPM | 0.8708 | 0.8774 | 0.8876 | 0.8996 | 0.8930 | 0.8934 |
| | w/o ESFE | 0.8803 | 0.8757 | 0.8806 | 0.9006 | 0.8937 | 0.8946 |
| | **LinkThief** | 0.9067 | 0.9077 | 0.9092 | 0.9226 | 0.9159 | 0.9214 |
| **Target Dataset** | | Brazil | Brazil | Europe | Europe | USA | USA |
| **Shadow Dataset** | | Europe | USA | Brazil | USA | Brazil | Europe |
| **Method** | w/o BGG | 0.8596 | 0.8729 | 0.9018 | 0.8984 | 0.9501 | 0.9425 |
| | w/o ESPM | 0.8824 | 0.8799 | 0.9004 | 0.8976 | 0.9464 | 0.9379 |
| | w/o ESFE | 0.7494 | 0.6283 | 0.8737 | 0.8045 | 0.9264 | 0.9127 |
| | **LinkThief** | 0.8881 | 0.8828 | 0.9038 | 0.9013 | 0.9522 | 0.9473 |

**Impact of Edge Subgraph Preparation Module:** We observe that without ESPM, LinkThief degrades the AUC score by about 3% on Citation dataset and by about 0.5% on airport dataset. This suggests that utilizing distinct subgraph sampling strategies for the target and shadow links benefits the attack model.

**Impact of Edge Structure Feature Extractor:** We find that without ESFE, LinkThief's AUC scores are reduced by about 2.5% on Citation dataset, about 3% on airport dataset, and even 20% in some cases. This indicates that the attack model benefits from using edge subgraph structure features as a complement to attack features.

### 5.4 Empirical verification of Prop.3.4

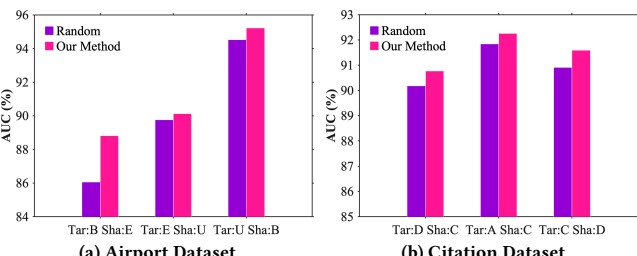

(a) Airport Dataset          (b) Citation Dataset

**Figure 4: Purple bars denote bridge building by randomly adding links, while pink bars represent our method which minimizes the representation distance. We use uppercase to represent datasets, e.g., B is Brazil.**

**Empirical study of Prop.3.4(1):** Prop.3.4 (1) suggests that the more similar the features of the shadow nodes and the target nodes in the edge subgraph, the more conducive to privacy theft. We use the bridge construction method that randomly adds edges to compare with the bridge construction method based on minimizing the distribution distance between the shadow node and the target node. As shown in Fig.4, compared with the former, the bridge graph constructed based on Prop.3.4 (1) has a higher AUC score in the former attacks. This proves the effectiveness of Prop.3.4 (1).

**Empirical study of Prop.3.4 (2):** Prop.3.4 (2) suggests that a larger proportion of target nodes in the edge subgraph is more conducive to privacy theft. Since the subgraph of the target link only samples target nodes, and the subgraph of the shadow link samples both target nodes and shadow nodes, the number of bridges is a measure of the number of target nodes in the shadow subgraph. In other words, the more bridges there are, the larger the proportion of target nodes in the shadow subgraph. As shown in Fig.5, with the

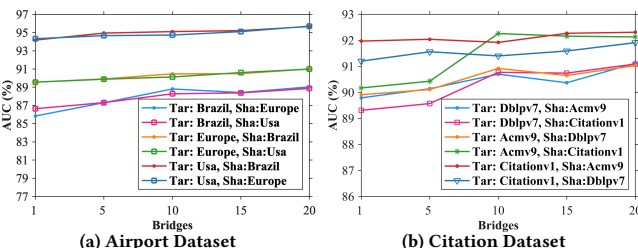

(a) Airport Dataset          (b) Citation Dataset

**Figure 5: The number of bridges indirectly reflects the proportion of the target node in the edge subgraph.**

increase in the number of bridges, the AUC scores of subsequent attacks exhibit an upward trend. This proves the effectiveness of it.

## 6 RELATED WORK

Link stealing attacks are a common type of membership inference attack on links, and several works have been proposed in this area. He et al.[15] are the first to propose the link stealing attack model. They propose eight link stealing attacks based on three types of background knowledge possessed by the attacker. The model was experimentally shown to be effective in stealing links between nodes. Based on this, Wu et al.[36] propose a privacy attack Link-Teller by considering the influence propagation in GNN training, which is a refinement of LSA-2 in [15]. Recently, Zhang et al.[43] propose a group-based attack paradigm that is able to solve the uneven vulnerability of GNNs by giving specific attack methods for different group characteristics, which is a refinement of LSA-0 in [15]. However, there is no further research on LSA-4 in [15], that is, the attacker's background knowledge includes partially leaked target graph and shadow graph. Moreover, they are all attacks based on the posterior similarity of nodes, which are not applicable to all links. Based on the progress of the current work, this paper proposes a link stealing attack model from a new perspective.

## 7 CONCLUSION

In this paper, we investigate the link stealing attack against links that are insensitive to similarity-based attacks and propose an improved attack method called LinkThief. We first empirically demonstrate the bottleneck of relying solely on node similarity as attack features, and then suggest that structure features of subgraph around links can be used as a complement to attack features. To obtain the edge subgraph structure features that span the target and shadow graphs, we introduce the concept of bridge graphs to connect the two graphs together. Through theoretical analysis, we summarize the criterion to measure the impact of the bridge and how to sample the subgraph around the target link and the shadow link, respectively. Based on the above findings, we design three modules for LinkThief to obtain edge subgraph structure features: Bridge Graph Generator (BGG), Edge Subgraph Preparation Module (ESPM), and Edge Structure Feature Extractor (ESFE). Finally, we input the attack features obtained by concatenating the structure features and similarity features into the attack model to obtain the link stealing results. Extensive experiments demonstrate the effectiveness of LinkThief. In future work, we will explore defenses against LinkThief.

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
