# OpenReview forum: "LinkThief: Combining Generalized Structure Knowledge with Node Similarity for Link Stealing Attack against GNN"
_acmmm.org/ACMMM/2024/Conference — MM2024 Poster_

### Official Review · Reviewer_RbVc · 2024-05-14

**Rating:** 4
**Confidence:** 3

**Summary:**

This paper focuses on improving link stealing attacks against GNNs for cases where the node posteriors might not be similar, which makes traditional similarity-based attacks ineffective. The main idea of authors is to combine node similarity with generalized structural features extracted from edge subgraphs around the sampled links to form attack features.

**Strengths:**

- Providing concrete evidence of traditional link stealing attacks’ bottleneck.
- Designing a constructive framework to combine node similarity with generalized structural features, supporting this framework with sufficient theoretical analysis as well.
  - In the analysis, authors answer RQ1, RQ2 with mathematically derivations and use following persuasive results to design their BGG, ESPM modules of the pipeline in the paper.
  - Explicitly discussing the roles of BGG, ESPM, and ESPE, combining with RL Policy gradient methods, contrastive learning in an innovative way.
- The experiments are complete, illustrating that LinkThief attack’s performance is prominent enough.
- The paper is well-organized and clearly written.

**Limitations:**

- In the GNN Defense-Attack field, defensing should be more crucial than attacking to some extent. It is acceptable that paper focus on one attack method, but some assumptions and predictions of the potential defense methods against authors’ own attacks are also expected, while authors do not provide any of their thoughts.
- There are typos in Eq.8.
- Some detailed explanations are still needed, such as why ω and ϕ share the same learning rate, why here only GCN is considered as the target model and shadow model architecture, and how to decide to build BGG with a 2-layer GCN, etc.
- According to the main experiment results, the attack performances on social networks are more notable, thus Twitch and Facebook dataset might be better for ablation study.
- Hope to see additional analysis on the computational efficiency, memory costs, and scalability of LinkThief compared to baselines as tradeoffs to utilize more graph features.
- The paper is novel and interesting, but its significance might not be much.

**Suitability:**

2

---

### Official Review · Reviewer_7TYK · 2024-05-25

**Rating:** 4
**Confidence:** 3

**Summary:**

The paper proposes a link stealing attack method called LinkThief, which combines generalized structure knowledge with node similarity to address the shortcomings of existing link stealing attack methods in GNNs. The study demonstrates that LinkThief can effectively steal links without additional assumptions, and provides extensive experimental results to illustrate the method's effectiveness.

**Strengths:**

1. The article is well-structured, providing a detailed background and an accurate abstract.

2. The method description is reasonable and logically clear.

3. The experiments are detailed and diverse, making the experimental section comprehensive.

**Limitations:**

1. The method comprises many modules, making the model structure rather complex.
2. Given the numerous modules, the article lacks an analysis of the computational resources and time costs associated with the method.

**Suitability:**

2

---

### Official Review · Reviewer_9Agg · 2024-05-25

**Rating:** 4
**Confidence:** 1

**Summary:**

This paper proposes LinkThief, an improved link stealing attack that combines generalized structure knowledge with node similarity, in a scenario where the attackers’ background knowledge contains partially leaked target graph and shadow graph.

**Strengths:**

Through theoretical analysis, the authors explore how to implement the aforementioned idea. On this basis, they propose LinkThief, an improved link stealing attack that comprises three modules to extract generalized structure features of edge subgraphs around links as supplementary for the attack model.

**Limitations:**

This is not my research direction. I have no more questions.

**Suitability:**

2

---

### Official Review · Reviewer_vGz5 · 2024-05-26

**Rating:** 5
**Confidence:** 3

**Summary:**

This paper proposes a novel link stealing attack method, LinkThief. Link stealing attack aims to identify the missing edges on the target graph, given a black box GNN model trained on the target graph, and a small portion of target graph edges (partial target graph). Previous methods utilize the similarity between two nodes to predict the existence of an edge between them. The paper points out the deficiency of only relying on node similarity by an intuitive visualization. Then, it proposes a method that combines the node similarity method and generalized structure knowledge. Since the target graph information is scarce, it introduces a shadow graph from the same domain to which we have full access. The first module, bridge graph generator, uses a GNN to help construct a bridge graph that links the shadow graph and the partial target graph, where the edges between the shadow graph and partial target graph aim to connect similar nodes. The second module, edge subgraph preparation, generates an edge subgraph based on the neighborhood of the two nodes. The third module, edge structure feature extractor, conducts contrastive learning between the edge subgraph and a similarity-preserving graph to ensure that nodes in the node subgraph capture the node similarity knowledge. Then, the node embeddings are combined with similarity measurements as features to input into the link prediction MLP.

**Strengths:**

(1)   The motivation is interesting and meaningful. Considering both structure information and similarity measurement is important.

(2)   The visualization of the previous method’s drawback is good and helps understanding.

(3)   The proposed model is well-designed. It has three modules, and it is supported by theoretical analysis.

(4)   The experiment results verify its effectiveness, where the proposed method clearly outperforms the baseline methods. The ablation studies clearly verify that each component in the model is important.

**Limitations:**

(1)   Limited model selection in the experiments. This paper only selects GCN as the shallow model and the target model in the experiments. It is worth studying whether the method also works on more complex GNN models, like GAT.

(2)   The writing can be improved. First, the introduction is a little confusing. Some terminologies appear for the first time in the introduction without explanation, such as “shallow graph” and “edge subgraph”. I have to refer to later sections to understand these terminologies. It is better to explain a terminology at its first appearance. Second, maybe the authors can consider reducing some words in the theoretical analysis (Section 3.4) and increasing some explanations in the methods section (Section 4). The theoretical results (Prop. 3.4) seem very intuitive to me and might not need so much analysis, and the methods might need more space to be fully explained. The authors can also consider not listening to this suggestion.

(3)   A small issue that does not impact my rating: you should replace the placeholders in “CCS Concepts” on the first page with the actual CCS concepts.

**Suitability:**

2

---

### Meta-Review · Area_Chair_Vobu · 2024-07-03

**Recommendation:** Accept (Poster)
**Confidence:** 5

**Metareview:**

The motivation of this study is interesting with the consideration of generalized structural information and similarity measurement. The limitation of previous studies on conventional link stealing attack approaches is clearly illustrated. The proposed model design of LinkThief with 3 modules is supported by theoretical analysis.